# Effectiveness of Elexacaftor/Tezacaftor/Ivacaftor Therapy in Three Subjects with the Cystic Fibrosis Genotype Phe508del/Unknown and Advanced Lung Disease

**DOI:** 10.3390/genes12081178

**Published:** 2021-07-29

**Authors:** Vito Terlizzi, Carmela Colangelo, Giovanni Marsicovetere, Michele D’Andria, Michela Francalanci, Diletta Innocenti, Eleonora Masi, Angelo Avarello, Giovanni Taccetti, Felice Amato, Marika Comegna, Giuseppe Castaldo, Donatello Salvatore

**Affiliations:** 1Cystic Fibrosis Regional Reference Center, Department of Paediatric Medicine, Anna Meyer Children’s University, 50139 Florence, Italy; michela.francalanci@meyer.it (M.F.); diletta.innocenti@meyer.it (D.I.); eleonora.masi@meyer.it (E.M.); giovanni.taccetti@meyer.it (G.T.); 2Cystic Fibrosis Center, AOR Ospedale San Carlo, 19104 Potenza, Italy; c.colangelo@tiscali.it (C.C.); marsicogio76@gmail.com (G.M.); micheledandria@gmail.com (M.D.); saverdon@gmail.com (D.S.); 3Infectious and Tropical Diseases Unit, Azienda Ospedaliero-Universitaria Careggi, 50134 Florence, Italy; avarelloa@aou-careggi.toscana.it; 4Department of Molecular Medicine and Medical Biotechnology, University of Naples, 20122 Naples, Italy; felice.amato@unina.it (F.A.); marika.comegna@unina.it (M.C.); giuseppe.castaldo@unina.it (G.C.); 5CEINGE—Advanced Biotechnology, 20122 Naples, Italy

**Keywords:** elexacaftor/tezacaftor/ivacaftor, cystic fibrosis, CFTR, nasal brushing, sweat chloride

## Abstract

We evaluated the effectiveness and safety of elexacaftor/tezacaftor/ivacaftor (ELX/TEZ/IVA) in three subjects carrying the *Phe508del*/unknown CFTR genotype. An ex vivo analysis on nasal epithelial cells (NEC) indicated a significant improvement of CFTR gating activity after the treatment. Three patients were enrolled in an ELX/TEZ/IVA managed-access program, including subjects with the highest percent predicted Forced Expiratory Volume in the 1st second (ppFEV_1_) < 40 in the preceding 3 months. Data were collected at baseline and after 8, 12 and 24 weeks of follow-up during treatment. All patients showed a considerable decrease of sweat chloride (i.e., meanly about 60 mmol/L as compared to baseline), relevant improvement of ppFEV_1_ (i.e., >8) and six-minute walk test, and an increase in body mass index after the first 8 weeks of treatment. No pulmonary exacerbations occurred during the 24 weeks of treatment and all domains of the CF Questionnaire-Revised improved. No safety concerns related to the treatment occurred. This study demonstrates the benefit from the ELX/TEZ/IVA treatment in patients with CF with the *Phe508del* and one unidentified CFTR variant. The preliminary ex vivo analysis of the drug response on NEC helps to predict the in vivo therapeutic endpoints.

## 1. Introduction

Cystic fibrosis (CF) is an autosomal recessive genetic disease due to mutations in the cystic fibrosis transmembrane conductance regulator (CFTR) gene [1]. To date, 360 CFTR variants are known to be CF-causing (https://cftr2.org/, accessed on 6 July 2021). These variants are classified according to the impact they have on the synthesis, processing or function of the CFTR protein. Progress in drug development has been substantial over the past decade, and new drugs that target the basic defect in CF have provided hope for patients. The first class of drugs to be successfully developed were CFTR potentiators, such as *ivacaftor*, which are small molecules that interact with the mutant channel to augment its opening probability, enhancing anion flux across the plasma membrane. At present, ivacaftor (Kalydeco^®^) is approved in the USA for children with CF aged four months or older who have at least one responsive variant in the CFTR gene, based on clinical and/or in vitro assay data. Kalydeco is also approved on its own in the EU for the treatment of patients with CF aged four months and older with gating variants [2,3,4].

CFTR correctors (i.e., *lumacaftor*, *tezacaftor*, *elexacaftor*) correct the processing and trafficking defects of the *Phe508del-CFTR* protein to enable it to reach the cell surface. The combination of a single corrector, both *lumacaftor* and *tezacaftor*, with the potentiator *ivacaftor* improves clinical outcomes including lung function and the rate of pulmonary exacerbations (PEx) in selected subgroups of patients with CF [5,6,7]. The use of the dual combination *lumacaftor/ivacaftor* (Orkambi^®^) is approved in the USA and EU for patients with CF 2 years and older who are homozygous for the *Phe508del* variant. Moreover, in the USA the combination *tezacaftor/ivacaftor* (Symdeko^®^) is a prescription treatment for patients aged 6 years and older who have two copies of the *Phe508del* variant, or who have at least one of the 154 variants responsive to treatment with *tezacaftor/ivacaftor*. In the EU *tezacaftor/ivacaftor* (Symkevi^®^) is used in patients with CF aged 6 years and above who have two copies of the *Phe508del* variant. It is also used in patients who have inherited the *Phe508del* variant from one parent as well as a “residual function” variant. The newest CFTR modulator is elexacaftor/tezacaftor/ivacaftor (ELX/TEZ/IVA) [8]. In the USA ELX/TEZ/IVA (Trikafta^®^) is a prescription medicine used for the treatment of CF in patients aged 6 years and older who have at least one copy of the *Phe508del* variant in the CFTR gene or another of 177 variants that are responsive in vitro to treatment with this triple combination. In the EU, the combination ELX/TEZ/IVA (Kaftrio^®^) is used in patients aged 12 years and older in which CF is due to at least one *Phe508del* variant in the CFTR gene. The approval was obtained thanks to fast and relevant improvements in lung function, the rate of PEx, sweat chloride concentration (SCC), CF Questionnaire-Revised (CFQ-R) respiratory domain scores and Body Mass Index (BMI) [8,9]. The triple combination helps the CFTR protein perform better than other modulators for an even greater number of patients with CF. Nevertheless, further research to extend the benefit of CFTR modulation to patients with responsive mutations other than the *Phe508del* variant or with rare genetic profiles is imperative.

Here we focus on patients with CF who carry the *Phe508del* variant and another unidentified variant even after extensive gene scanning. This subgroup of patients (*Phe508del*/unknown), not included in clinical trials so far, is quite numerous in Italy (i.e., 71 patients, about 1.3% of the overall Italian CF patients), and some of them suffer from severe lung disease. We report the clinical data of the first three adult patients with CF and advanced lung disease, carrying the *Phe508del*/unknown CFTR genotype treated with ELX/TEZ/IVA. The ex vivo analysis of nasal epithelial cells from such patients predicted a response to the three drugs [10], and this result allowed for the possibility to be treated in the clinical setting. We now describe the clinical response to the treatment and focus on the emerging problem of the right to access innovative treatments for all potentially eligible patients with CF.

## 2. Materials and Methods

This is a retrospective, observational study involving three patients with CF, followed at the CF centers of Florence and Potenza, Italy, carrying the *Phe508del*/unknown CFTR genotype after gene scanning (detection rate 98%) and multiplex ligation-dependent probe amplification (MLPA) [11,12]. The patients were enrolled in an ELX/TEZ/IVA managed access program (MAP), performed between October 2019 and April 2021 in Italy for patients with CF aged 12 years and older, heterozygous for the *Phe508del* variant and a *minimal function* (MF) variant and with severe lung disease, defined as either a highest predicted Forced Expiratory Volume in the 1st second (FEV_1_) < 40% in the preceding 3 months or being on a lung transplant waiting list. No previous CFTR modulators had been prescribed.

In all three patients, the ex vivo analysis of nasal epithelial cells had predicted a good response to ELX/TEZ/IVA treatment [10]. The patients were individually approved by the drug manufacturer (Vertex Pharmaceuticals Inc., Boston, MA, USA).

ELX/TEZ/IVA was administered orally at a dose of two tablets, each containing 100 mg ELX/50 mg TEZ/75 mg IVA, in the morning and one tablet, containing 150 mg IVA, in the evening with fatty meals. Throughout the study, all patients continued to take their pre-study medications.

The main outcome measures included FEV_1_ (calculated according to the Global Lung Function Initiative (GLI) [13], body mass index (BMI) (the weight in kilograms divided by the square of the height in meters, kg/m^2^), sweat chloride concentration (SCC) (mmol/L), six-minute walk test (6MWT, m) distance, rate of PEx, CFQ-R, and treatment-related adverse events following ELX/TEZ/IVA treatment.

PEx was defined as a change in antibiotic therapy (intravenous (IV), inhaled, or oral) for any 4 or more of the following signs/symptoms: change in sputum; new or increased hemoptysis; increased cough; increased dyspnea; malaise, fatigue, lethargy; temperature above 38 °C; anorexia or weight loss; sinus pain or tenderness; change in sinus discharge; change in physical examination of the chest; decrease in pulmonary function by 10%; or radiographic changes indicative of pulmonary infection [2].

The CFQ-R was administered at each encounter. For the CFQ-R respiratory domain score (range: 0 to 100, with higher scores indicating a higher patient-reported quality of life with regard to respiratory symptoms), the minimum clinically important difference was 4 points [14].

Clinical data were collected at baseline and 8, 12 and 24 weeks after starting ELX/TEZ/IVA treatment. The study protocol (project identification code VX18-445-902) conformed to the amended Declaration of Helsinki and all participants provided written informed consent. The collection of data for each patient was an integral part of the approval by each local Ethics Committee (Paediatric Ethic committee of Anna Meyer Children’s Hospital, Florence, Italy, on 17 December 2019 for patient 1 and Ethic committee of Basilicata Region on 7 August 2020 for patients 2 and 3) of the compassionate use program.

## 3. Results

### 3.1. Clinical Characteristics of the Enrolled Patients

Patient 1 is now a 48-year-old Caucasian woman, diagnosed as having CF with pancreatic insufficiency (fecal elastase < 100 µg/gr) at the age of 8 years. Her CF was characterised by persistent productive cough, diarrhea and pathological SCC (i.e., 104–102 mmol/L). At the start date of the ELX/TEZ/IVA treatment she suffered from chronic respiratory failure (ppFEV_1_ < 30 for at least 6 months and needing long-term therapy with oxygen), such that she had been included in the waiting list for lung transplantation. Furthermore, she had chronic infection by *Pseudomonas aeruginosa* (Pa) and by *Burkholderia gladioli*, insulin dependent CF-related diabetes and non-cirrhotic CF-related liver disease.

Patient 2 is now a 59-year-old Caucasian woman, diagnosed as having CF with pancreatic sufficiency at the age of 20 years, on the basis of chronic productive cough, evidence of bronchiectasis at CT scan and infection by Pa. The SCC at diagnosis was pathological (i.e., 114–117 mmol/L). The clinical course of the lung disease progressively worsened until the development of chronic respiratory failure and acquisition of multi-resistant Pa [15]. The subject initiated long-term therapy with oxygen at the age of 57 years; in the last year ppFEV_1_ declined to 26, and she experienced six episodes of PEx requiring treatment with IV antibiotics.

Patient 3 is now a 43-year-old Caucasian woman, diagnosed as having CF with pancreatic sufficiency at the age of 18.3 years, on the basis of familiarity (one brother was also diagnosed with CF), chronic productive cough, evidence of bronchiectasis at CT scan and infection by Pa. The SCC at diagnosis was pathological (i.e., 66–71 mmol/L). The clinical course of the lung disease was progressively worsening until the development of chronic respiratory failure and acquisition of multi-resistant Pa [15]. The subject initiated therapy with nocturnal oxygen at the age of 42 years; in the last year ppFEV_1_ declined to 29, and she experienced seven episodes of PEx requiring treatment with IV antibiotics.

### 3.2. Outcome Measures Following Elexacaftor/Tezacaftor/Ivacaftor Treatment

The main outcome measures following ELX/TEZ/IVA treatment are reported in Table 1. The SCC decreased progressively in all patients with reductions higher than 60 mmol/L, reaching values within the normal range, at 24 weeks of treatment.

Furthermore, the treatment resulted in relevant improvements of the ppFEV_1_ (Table 1). Subject #1 showed a difference of 13, 18 and 15 points at week 8, 12 and 24, respectively. Subject #2 showed a difference of 8.4, 8.2 and 6.2 as compared to baseline at week 8, 12 and 24, respectively. Subject #3 also improved lung function, with a 20% increase of ppFEV_1_ at week 8 and 24.5% at week 12 and 24.

In addition, physical performance improved following treatment with ELX/TEZ/IVA. The walked distance of subject #1 increased from 360 m at baseline to 558 m after 8 weeks and was sustained after 12 and 24 weeks of treatment. Similarly, subject #2 walked over 80 m longer at 24 weeks. Finally, patient #3 also improved walking distance by 72 m as compared to baseline.

BMI improved in turn in all the cases, with a difference of 1.34 in subject #1, 1.4 in subject #2, and 3.1 in subject 3, relative to baseline, after 24 weeks of treatment.

The CFQ-R respiratory domain score improved significantly during the 6 months of triple combination treatment. The score at baseline was 44.4 in subjects #1 and #3 and 50 in subject #2, respectively. After 8 weeks, the respiratory domain score improved in all the patients. The improvement was sustained at 12 weeks and 24 weeks. Further domains of the CFQ-R also showed an overall improvement in quality of life (Table 1).

Finally, subject #1 halved their daily insulin dose after the first 4 weeks of treatment and discontinued oral steroid therapy in the third month. Furthermore, mean oxygen saturation during sleep changed from 88% to 93%, and she discontinued daytime oxygen therapy after 8 weeks. Subject #2 discontinued oxygen therapy during the daylight hours after 4 weeks of treatment. Subject #3 demonstrated improved nocturnal hypoxemia (mean oxygen saturation during sleep changed from 89% to 94%) and discontinued oxygen after 12 weeks of treatment.

Furthermore, treatment with ELX/TEZ/IVA reduced the requirement of antibiotic therapy. During the 3 months before the commencement of the ELX/TEZ/IVA therapy, all subjects experienced continual instances of PEx, requiring treatment with IV antibiotics and continuous oral antibiotic therapy between courses of IV therapy. Treatment with the triple combination resulted in no need for antibiotic therapy in the first 24 weeks of follow-up for all three patients. No changes about the microbiological status were reported.

No adverse events leading to interruption of ELX/TEZ/IVA therapy were reported, and no patients withdrew from the treatment. No relevant abnormal results attributable to ELX/TEZ/IVA were identified in clinical laboratory tests (serum chemistry, hematology, coagulation, liver function tests, and urinalysis), vital signs, or physical examinations.

## 4. Discussion

This is the first study that evaluated the clinical effectiveness of the treatment with ELX/TEZ/IVA in patients with CF carrying the *Phe508del*/unknown CFTR genotype. The prerequisites of the treatment were severe clinical phenotype and the preliminary ex vivo testing on nasal epithelial cells that had demonstrated a severe impairment of CFTR activity in untreated cells, improved by ELX/TEZ/IVA [10].

The main efficacy outcomes of the treatment were the SCC and the ppFEV_1_. SCC is the main biomarker of efficacy in terms of improvement of CFTR protein expression/function induced by drugs [16,17,18]. We observed a decrease of the SCC in all three subjects already after 8 weeks of treatment. The reduction of SCC was impressive, i.e., more than 60 mmol/L, and was more relevant than that reported in patients with CF carrying a *Phe508del*/MF genotype treated with ELX/TEZ/IVA during the phase III trial [8].

Furthermore, the treatment with ELX/TEZ/IVA of our patients resulted in a relevant improvement in the absolute ppFEV_1_, similar to that reported by the clinical trial [8]. We also found that the treatment improved the 6MWT distance, a simple and well-tolerated test that reflects activities of daily living [19]. The improvement in the walked distances of our subjects was clinically relevant, higher than the mean of 54 m previously reported to be associated with a noticeable clinical difference in patients with stable chronic obstructive pulmonary disease [20].

The effect of the triple combination therapy on healthcare utilization is another clinically important endpoint in our study. In the first six months of follow-up after initiating ELX/TEZ/IVA all three patients did not need antibiotic therapy, whereas continuous treatments with oral or IV antibiotics had been necessary in the three months before starting the treatment. This result is comparable to that obtained in the phase-III trial (rate of PEx of 63% lower with respect to placebo) [8]. Furthermore, improvements in weight and BMI are well-recognized benefits of CFTR modulator therapy [21,22] but the effect can be variable depending on the CFTR modulator used. In our cases BMI also improved in all the subjects as compared to baseline. Finally, the improvement of lung function and BMI corresponded with the patients’ reported outcomes of quality of life, as expressed by the improvement of the domains of CFQ-R.

We acknowledge limitations of our study, i.e., that it was conducted with only three cases and for a short follow-up period. Nonetheless, the fast response to the treatment with ELX/TEZ/IVA and the sudden trend inversion of the clinical status in subjects with a very severe lung involvement and with only a known CFTR mutation warrant consideration.

The development of CFTR modulators is changing the natural history of CF. About 90% of US patients with CF carry one or two copies of the *Phe508del* CFTR allele, thus making almost all patients eligible for the new drugs [23]. The frequency of the *Phe508del* is variable in other countries, with a decreasing prevalence from Northwest to Southeast Europe [24,25]. Data from the Italian CF Registry (ICFR) show that only 20.9% of the registered patients are homozygous for *Phe508del*, and only 48.8% have at least one *Phe508del* allele [26]. Furthermore, despite the wide availability of genome sequencing technologies, 71 patients with CF (1.3% of the Italian patients) have a genotype *Phe508del*/unknown [26]. Similar data comes from other countries of Southern Europe, where 1.0–1.5% of patients with CF have the *Phe508del* mutation with one unknown mutation *in-trans* [27].

Our study shows that these patients could also significantly benefit from the treatment with ELX/TEZ/IVA, according to the indication of eligibility for subjects with CF with genotypes *Phe508del*/any. Of course, for these patients it is mandatory that diagnostic criteria of CF are fulfilled and that direct measures demonstrate missing CFTR activity corrected by ex vivo drug screen to predict the best individual treatment. On the other hand, given the high number of CFTR variants, functional drug testing on ex vivo models from patients represents a major step to predict effective treatments in an individual setting. International projects to evaluate the efficacy of new molecules on organoids from intestinal or nasal cells from patients with CF with rare mutations are ongoing, but such models are invasive and expensive [28,29,30]. The model of human nasal epithelial cells is a rapid, minimally invasive and effective tool to investigate the effect of novel variants and to assess the effect of novel molecular therapies in individual patients [30,31,32,33,34]. Indeed, we recently reported on treatment with *ivacaftor* of a child with a rare CFTR variant, on the basis of the demonstration of the positive effect on the CFTR gating activity recorded on nasal epithelial cells [34]. Therefore, drug response obtained in ex vivo testing correlates with changes in in vivo therapeutic endpoints [33].

## 5. Conclusions

The present case series further confirms that the response predicted on nasal epithelial cells correlates with in vivo therapeutic endpoints, such as SCC values, confirming the efficacy of such model as a predictor of clinical effectiveness of novel drugs. Moreover, it also opens a new possibility of treatment for patients with CF with a *Phe508del* and one unidentified CFTR variant.

## Figures and Tables

**Table 1 genes-12-01178-t001:** Comparison of change in all variables over the treatment period.

Characteristics	Baseline	After 8 Weeks	After 12 Weeks	After 24 Weeks	Baseline	After 8 Weeks	After 12 Weeks	After 24 Weeks	Baseline	After 8 Weeks	After 12 Weeks	After 24 Weeks
	Subject 1				Subject 2				Subject 3			
Sweat chloride (mEq/L)	110	26	46	46	103	40	34	31	85	48	30	29
ppFEV_1_ (%)	26.0	39.0	44.0	41.0	25.8	34.2	34.0	32.0	29.0	48.9	53.5	53.6
BMI (Kg/m^2^)	20.3	20.9	21.6	21.2	24.4	25.8	25.7	26.2	19.5	20.5	21.6	22.6
6MWT (m)	360	558	545	521	403	480	495	485	500	544	560	572
Oxigen saturation	84	89	96	98	75	78	81	91	82	89	93	95
Heart rate	138	139	120	110	119	129	145	122	154	141	130	135
CFQ-R 14 + domain												
Physical functioning	16.67	62.50	62.50	58.33	25.00	41.67	100.00	58.33	12.50	62.50	75.00	100.00
Role perception	91.67	91.67	100.00	91.67	66.67	83.33	100.00	83.33	58.33	83.33	83.33	100.00
Vitality	41.67	83.33	83.33	75.00	25.00	50.00	83.33	75.00	25.00	75.00	75.00	83.33
Emotion	73.33	100.00	100.00	93.33	40.00	46.67	46.67	66.67	46.67	66.67	60.00	73.33
Social perception	72.22	83.33	88.89	77.78	22.22	50.00	38.89	44.44	50.00	55.56	77.78	88.89
Body image	88.89	100.00	100.00	100.00	100.00	55.56	100.00	100.00	66.67	77.78	88.9	100.00
Eating disturbance	100.00	100.00	100.00	100.00	55.56	55.56	100.00	66.67	55.56	100.00	100.00	100.00
Treatment burden	33.33	44.44	55.56	55.56	77.78	77.78	77.78	77.78	22.22	55.56	66.67	77.78
Health perception	55.56	66.67	66.67	55.56	66.67	88.89	88.89	100.00	11.11	66.67	66.67	88.89
Weight	66.67	100.00	100.00	100.00	100.00	66.67	66.67	100.00	33.33	100.00	100.00	100.00
Respiratory symptoms	44.44	83.33	83.33	83.33	50.00	72.22	83.33	77.78	44.44	83.33	83.33	83.33
Digestive symptoms	88.89	100.00	100.00	88.89	88.89	77.78	88.89	77.78	88.89	100.00	77.78	100.00

## Data Availability

All data are available by contacting the corresponding author.

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
