# Peer review of "Effectiveness of Elexacaftor/Tezacaftor/Ivacaftor Therapy in Three Subjects with the Cystic Fibrosis Genotype Phe508del/Unknown and Advanced Lung Disease"

_genes, 2021, doi:10.3390/genes12081178_

Round 1

Reviewer 1 Report

Overall comments:

Comegna et al. reported a case series of three cystic fibrosis patients with F508del/unknown genotype successfully treated with Elexacaftor/Tezacaftor/Ivacaftor (ELX/TEZ/IVA). All the three patients studied exhibited significant improvement of skin chloride concentration, fecal elastase concentration, respiratory function, physical performance, infection frequency, BMI, and multiple domains of CFQ-R and so on.

Although the number of subjects studied was limited, the findings shown in this paper, coupled with the authors’ recently published in vitro paper,  is an encouraging and important clinical outcome data of ELX/TEZ/IVA for CF patients with F508del/unknown mutation genotypes.

The reviewer suggests the journal publish this paper after a minor revision.

Minor comments:

Line 49: TRILAFTA was in all upper letters while the other drugs were not.

Line 80: The patients’ previous medication details (especially CFTR corrector/potentiators if any) should be clarified.  

Table 1: mSatO2 and mHR should not be abbreviated or full term should be clarified in the main text or the legend.

Author Response

EFFECTIVENESS OF ELEXACAFTOR/TEZACAFTOR/IVACAFTOR THERAPY IN THREE SUBJECTS WITH THE CYSTIC FI-BROSIS GENOTYPE PHE508DEL/UNKNOWN AND ADVANCED LUNG DISEASE

Terlizzi V et al.

Reviewer 1

Overall comments:

Comegna et al. reported a case series of three cystic fibrosis patients with F508del/unknown genotype successfully treated with Elexacaftor/Tezacaftor/Ivacaftor (ELX/TEZ/IVA). All the three patients studied exhibited significant improvement of skin chloride concentration, fecal elastase concentration, respiratory function, physical performance, infection frequency, BMI, and multiple domains of CFQ-R and so on.

Although the number of subjects studied was limited, the findings shown in this paper, coupled with the authors’ recently published in vitro paper, is an encouraging and important clinical outcome data of ELX/TEZ/IVA for CF patients with F508del/unknown mutation genotypes.

The reviewer suggests the journal publish this paper after a minor revision.

Re: We thank the reviewer for the positive comment.

Minor comments:

Line 49: TRILAFTA was in all upper letters while the other drugs were not.

Re: we have modified the sentence according to other drugs.

Line 80: The patients’ previous medication details (especially CFTR corrector/potentiators if any) should be clarified.  

Re: no patient had previously been treated with CFTR modulators. We have explained this point in methods.

Table 1: mSatO2 and mHR should not be abbreviated or full term should be clarified in the main text or the legend.

Re: we have changed in mSat02 in oxygen saturation and HR in heart rate (in table 1).

Reviewer 2

The submitted work is a very important contribution. There are still quite a number of patients who present the clinical picture of cystic fibrosis but whose molecular genetic background could only be partially elucidated. This was also the case in the patients studied here. In this case, all three had F508del as one CFTR variant, while the second CFTR variant remained unclear despite extensive genetic analysis. However, since the chloride concentration measured in sweat was pathological in all three patients, the diagnosis is considered confirmed. In this respect it was only logical to try the triple therapy with ELX/TEZ/IVA due to the severity of the CF disease and to evaluate the patients concomitantly. 

Re: We thank the reviewer for the positive comment.

Minor points:

It would have been interesting to also record the microbiology in sputum or throat swabs, because we had found in our patients with F508del/F508del and F508del/MF that the microbiology changed relatively rapidly for the better. Perhaps the authors could add this for completion.

Re: There were no post-Trikafta changes regarding the microbiological status. We have added this sentence in results (section 3.2).

The oxygen saturation as a baseline value was so low that I assume that this is the value measured without oxygen supplementation. In patients #1 and #3, it looks like it might now be possible to do without it. If so, it would be interesting to know if the oxygen concentrations were measured only while awake or also during sleep. It would not be a problem if it had been measured only while awake, the results are impressive even so. But for the sake of completeness, it should be noted.

We: patient 1 and 3 improved the nocturnal hypoxemia. We have added these data in results.

Reviewer 2 Report

The submitted work is a very important contribution. There are still quite a number of patients who present the clinical picture of cystic fibrosis but whose molecular genetic background could only be partially elucidated. This was also the case in the patients studied here. In this case, all three had F508del as one CFTR variant, while the second CFTR variant remained unclear despite extensive genetic analysis. However, since the chloride concentration measured in sweat was pathological in all three patients, the diagnosis is considered confirmed. In this respect it was only logical to try the triple therapy with ELX/TEZ/IVA due to the severity of the CF disease and to evaluate the patients concomitantly. 

Minor points:

It would have been interesting to also record the microbiology in sputum or throat swabs, because we had found in our patients with F508del/F508del and F508del/MF that the microbiology changed relatively rapidly for the better. Perhaps the authors could add this for completion.

The oxygen saturation as a baseline value was so low that I assume that this is the value measured without oxygen supplementation. In patients #1 and #3, it looks like it might now be possible to do without it. If so, it would be interesting to know if the oxygen concentrations were measured only while awake or also during sleep. It would not be a problem if it had been measured only while awake, the results are impressive even so. But for the sake of completeness, it should be noted.

Author Response

(The authors gave the same response as above.)
